# Evaluation of the ActiMotus Software to Accurately Classify Postures and Movements in Children Aged 3–14

**DOI:** 10.3390/s24206705

**Published:** 2024-10-18

**Authors:** Charlotte Lund Rasmussen, Danica Hendry, George Thomas, Amber Beynon, Sarah Michelle Stearne, Juliana Zabatiero, Paul Davey, Jon Roslyng Larsen, Andrew Lloyd Rohl, Leon Straker, Amity Campbell

**Affiliations:** 1School of Allied Health, Curtin University, Perth, WA 6102, Australia; danica.hendry@curtin.edu.au (D.H.); george.thomas1@uq.edu.au (G.T.); amber.beynon@curtin.edu.au (A.B.); sarah.stearne@curtin.edu.au (S.M.S.); juliana.zabatiero@curtin.edu.au (J.Z.); l.straker@curtin.edu.au (L.S.); a.campbell@curtin.edu.au (A.C.); 2ARC Centre of Excellence for the Digital Child, Brisbane, ACT 2609, Australia; andrew.rohl@curtin.edu.au; 3Health and Wellbeing Centre for Research Innovation, School of Human Movement and Nutrition Sciences, The University of Queensland, Brisbane, QLD 4006, Australia; 4School of Nursing, Curtin University, Perth, WA 6102, Australia; p.davey@curtin.edu.au; 5The National Research Centre for the Working Environment, 2100 Copenhagen, Denmark; xjla@nfa.dk; 6School of Electrical Engineering, Computing and Mathematical Sciences, Curtin University, Perth, WA 6102, Australia; 7Curtin Institute for Data Science, Curtin University, Perth, WA 6102, Australia

**Keywords:** accelerometry, validation, physical activity, sedentary behaviour

## Abstract

Background: ActiMotus, a thigh-accelerometer-based software used for the classification of postures and movements (PaMs), has shown high accuracy among adults and school-aged children; however, its accuracy among younger children and potential differences between sexes are unknown. This study aimed to evaluate the accuracy of ActiMotus to measure PaMs among children between 3 and 14 years and to assess if this was influenced by the sex or age of children. Method: Forty-eight children attended a structured ~1-hour data collection session at a laboratory. Thigh acceleration was measured using a SENS accelerometer, which was classified into nine PaMs using the ActiMotus software. Human-coded video recordings of the session provided the ground truth. Results: Based on both F1 scores and balanced accuracy, the highest levels of accuracy were found for lying, sitting, and standing (63.2–88.2%). For walking and running, accuracy measures ranged from 48.0 to 85.8%. The lowest accuracy was observed for classifying stair climbing. We found a higher accuracy for stair climbing among girls compared to boys and for older compared to younger age groups for walking, running, and stair climbing. Conclusions: ActiMotus could accurately detect lying, sitting, and standing among children. The software could be improved for classifying walking, running, and stair climbing, particularly among younger children.

## 1. Introduction

The time children spend in different postures and movements (e.g., sitting, standing, and walking) is well-acknowledged to influence their health and development. For example, children spending extensive time sitting may be at increased risk of obesity and metabolic syndrome [1,2]. In contrast, more dynamic movements such as walking and running are essential for healthy physical development, including bone health, muscular strength, and motor skill development [3,4]. Accordingly, obtaining valid measurements of children’s physical behaviours is a high priority for public health authorities [5].

Much of the current evidence on the health and development implications of children’s physical behaviours is based on proxy-reported measurements, such as questionnaires, interviews, or diaries. However, these methods are prone to measurement error, both non-random bias (e.g., social desirability bias) and random error (e.g., recall uncertainty) [6,7]. Therefore, it has become increasingly popular to obtain device-based measurements of children’s physical behaviours using wearable sensors such as accelerometers [8].

Most of the previous accelerometer studies have summarised children’s daily physical behaviours using activity “counts” of the accelerometry outputs [9]. In brief, counts express the magnitude of acceleration of the body part they are attached to as measured by accelerometers in a specific time interval (i.e., epoch length) [10]. The amount of daily time spent in different physical behaviour intensity bands (e.g., sedentary, light intensity, or moderate-to-vigorous physical activity) can then be obtained by classifying the activity counts, based on cut-off points corresponding to predefined thresholds of the metabolic equivalent of tasks [11,12]. Although the use of accelerometer counts to assess daily physical behaviours enables greater accuracy compared to self-reported measurements [9], it does not provide information on specific postures within the same intensity band, such as sitting and standing. This distinction can be of particular importance as each posture may have discrete health implications [13]. Additionally, public health messages based on intensity (e.g., increase time spent in ”moderate” intensity) may be harder for parents and children to understand compared to posture and movement-based messages (e.g., increase time spent running). Thus, information on children’s daily postures and movements can potentially provide a better understanding of the influence on health and development and aid in developing better policies and public health messages [14].

Systems using data from a single sensor worn on the hip or wrist have been most commonly used in previous population research [14]; however, these systems have been relatively poor at differentiating between sitting and standing [15]. Instead, thigh-worn accelerometry has been shown to differentiate these postures with high levels of accuracy (>90%) among adults [16,17,18]. However, validation studies of postures and movements based on thigh-worn accelerometry among children are scarce [19]. Five studies have assessed the accuracy of a thigh-worn accelerometer for estimating postures and movements among children [20,21,22,23,24], of which only three have examined multiple postures and movements [20,21,24]. The studies by Stewart et al. and Brønd et al. used the Axivity AX3 (Axivity, Newcastle, UK) accelerometer and customised software (i.e., machine learning or decision tree algorithms). Stewart et al. validated their machine learning system on 42 children, aged 7–15 years, performing ten semi-structured activities in a laboratory-based setting, whilst Brønd et al. validated a decision tree algorithm on 96 children, aged 3–16 years, performing 11 structured activities in a school environment. Both studies found a thigh-worn accelerometer to provide highly accurate estimates of standing (>95% balanced accuracy), walking (>80% balanced accuracy), and running (>90% balanced accuracy). The study by Lendt et al. was based on the same dataset as Stewart et al. as well as a free-living dataset from 15 children aged 8–12 years who wore an Axivity AX3 and a wearable camera for a 2-h period in a free-living environment [24]. Based on these datasets, two decision tree algorithms were evaluated—the algorithm published by Brønd et al. and its precursor, the Acti4 algorithm [25]. The Acti4 algorithm is implemented in ActiMotus, the data processing algorithm that is used in Motus, which is a wearable thigh-worn accelerometry-based system developed for posture and movement surveillance among adults [26]. The original Acti4 algorithm has demonstrated high balanced accuracy for estimating common postures and movements among adults [24,25]. Lendt et al. found a good classification performance of both the original and modified algorithm for standing (≥90% balanced accuracy), walking (>89% balanced accuracy), and running (≥86% balanced accuracy), while the accuracy found for detecting lying was considered as moderate (≤62% balanced accuracy) [24]. This result was found across both laboratory and free-living conditions.

These studies make important contributions to our understanding of the validity of using thigh-worn accelerometry for capturing postures and movements among children. Nevertheless, Stewart et al. and Lendt et al. did not evaluate sensors on younger children (i.e., below 7 years) although postures and movements are likely to differ among younger compared to older children [27], and Brønd et al. did not evaluate the accuracy of lying. Additionally, neither study assessed stair climbing, which is likely to be important to measure when aiming to capture the full range of children’s daily movement behaviours. Finally, potential sex differences were not investigated in the previous studies, despite boys and girls potentially moving differently [28,29,30].

Thus, there is further need to develop and validate wearable sensor-based systems enabling measurements of a range of postures and movements among children, including young children. Obtaining these robust measurements is essential to enable evidence-based guidance to encourage appropriate physical activity in children to support their health and development. The Motus system was specifically designed to ensure ease of use and low burden for both researchers and participants and, thus, has the potential to be widely applicable to small- and large-scale intervention and observation studies. While evidence shows this system does provide a robust and highly useful system among adults, its capability for use among children is not yet clear. Accordingly, this study aimed to evaluate the criterion validity of ActiMotus version 1.1.0, the software used in Motus, to classify postures and movements among children between 3 and 14 years in a laboratory setting by comparing to human-coded video. We further assessed if the sex or age of children impacted the accuracy.

## 2. Materials and Methods

### 2.1. Design

An experimental laboratory study was conducted where children performed a set of activities designed to require a range of postures and movements whilst being video recorded and wearing a thigh-worn accelerometer.

### 2.2. Participants

Typically developing children between 3 and 14 years old were recruited using social and workplace networks via word of mouth, flyers, and advertisements on research-associated social media accounts. Participant recruitment aimed to be sex- and age-stratified, whereby approximately equal numbers of boys and girls for each age were recruited. Children were grouped into three age brackets according to Australian school ages: 3–5 years old to represent a pre-school population; 6–10 years old to represent a primary-school-aged population; and 11–14 years old to represent a secondary-school-aged population/early adolescence. 

Children were excluded if their caregiver indicated they had any psychological or physical clinical diagnosis that may have influenced their ability to understand and follow instructions or perform requested movements. Following completion of informed caregiver consent and age-appropriate child verbal or written assent, caregivers completed a brief sociodemographic questionnaire about their child. 

### 2.3. Protocol

Eligible children attended a single data collection session (~1 h) at a Curtin University laboratory. Children and their caregivers attended the session either by themselves (n = 10) or with a friend/sibling (n = 38). When children attended with a friend/sibling, data were collected from both children simultaneously. Two to three researchers with experience working with children were present at each session. Following laboratory familiarisation, the accelerometer was attached to the child (see details of sensor below). Children were given the opportunity to move around whilst wearing the accelerometer to familiarise themselves with the collection equipment.

The session began with a series of synchronisation trials. The children were asked to stand still for ~5 secs and then to perform 5 jumps. Children were then asked to engage in a range of different age-appropriate digital and non-digital activities, performed for 2–3 min each, aiming to elicit diverse postures and movements (see Appendix A). Postures and movements targeted were chosen to reflect the diverse movement vocabulary of children and potential implications for children’s health following input from community and advisory groups, including a group of parents and a group of paediatric health and education professionals. Where possible, tasks were performed in a standardised order. However, flexibility in the protocol was allowed to ensure participant enjoyment and adherence while maintaining optimal collection of all postures and movements.

### 2.4. Context Video

All tasks were video recorded using two iPhones (model A2403; Apple, Inc., Cupertino, CA, USA) recording at 30 Hz (superwide lens) positioned on either side of the laboratory on one-meter-tall tripods ~4–6 m from the child, to allow capture of the full space and be less intrusive for the participants. The recording that captured most of the laboratory was used as the primary video and the other was the secondary video. The primary video recording was manually annotated for “instances” of different postures and movements by a trained member of the research team. Specifically, a coder would watch the video and enter the start and end video frame number of each contiguous instance of a posture or movement. A total of 22 postures and movements were coded using the definitions in Table 1, along with two additional code options (unsure and uncodable). While the ActiMotus software has a “moving” category, this category was not coded as pilot testing showed that children fidgeted so much it was too difficult for humans to code reliably. The coded postures and movements were used as the ground truth for validation of the postures and movements classified by ActiMotus. The secondary video was used when the primary video did not provide an adequate view of the child, or its recording was unexpectedly interrupted (two sessions).

#### Inter-Rater Reliability of Video Coding

The videos from 3 participants (one from each age group) were independently coded by a second researcher to determine inter-rater reliability via assessing percentage agreement and Cohen’s Kappa. An almost perfect inter-rater reliability was observed for all age groups, with an average percentage agreement of 96.2% (range: 95.5–98.6%) and average Kappa of 0.90 (range: 0.84–0.93).

### 2.5. Accelerometer Hardware and Software

One SENS motion PLUS 12.5 (SENS Innovation ApS, Copenhagen, Denmark) accelerometer was secured to the anterior mid-thigh of each child using a medically approved patch [18,31]. The SENS accelerometer operated at 12.5 Hz, weighed 7 g, was 47 × 22 × 4.5 mm in size, and recorded accelerations in the *x-*, *y-*, and *z*-axes with a range of ±4 G.

A custom-developed software, ActiMotus version 1.1.0, was used to classify postures and movements based on 12.5 Hz data from the accelerometers. The ActiMotus software is based on the previously developed Acti4 [25] and ActiPASS software [32] and is publicly available at https://github.com/motus-nfa/ActiMotus [33]. ActiMotus uses accelerometer data resampled to 30 Hz in 2 s windows with 50% overlap to classify different postures and movements with a rule-based algorithm, which considers the inclination of the thigh, the angle of the thigh, the standard deviation of thigh accelerations, and the rotation of the thigh [25,32]. The postures and movements determined by ActiMotus are as follows: *lying*, *sitting*, *standing*, *moving* (upright postures that have more movement than standing still but less movement than walking), *walking*, *running*, *stair climbing*, and *cycling*. A final classification of *rowing* is used to describe periods in which higher intensity movements are performed that are not classified as running, stair climbing, or cycling. The final output from ActiMotus consisted of a 1 Hz data file listing the predicted activity, which was upsampled to 30 Hz frequency to allow for merging with the video codes file.

### 2.6. Statistical Analysis

The SENS accelerometer data were time-synchronised with the video codes using a custom-made Python [34] program to allow for the comparison of ActiMotus and video classifications. As the ActiMotus software was not developed to classify children-specific postures and movements, this analysis only evaluated how well the ActiMotus software classified the following “standard” postures and movements: *lying* (prone lying, supine lying, and side lying combined), *sitting*, *standing*, *walking*, *running,* and *stair climbing* (walking upstairs, walking downstairs, running upstairs, and running downstairs combined). Thus, accelerometer data that were human-coded as any other posture or movement were excluded from the analysis. Note that although the data fed to the ActiMotus algorithm were restricted in the current study, the algorithm could output postures and movements not in the input set, i.e., *moving*, *cycling*, and *rowing*.

Confusion matrices were generated to summarise classification accuracy and identify patterns of misclassification for each posture and movement. Evaluation of ActiMotus accuracy for classification of each posture and movement was performed by calculating sensitivity (true positives/[true positives + false negatives]), specificity (true negatives/[true negatives + false positives]), and balanced accuracy ([sensitivity + specificity]/2). To evaluate the overall accuracy of ActiMotus, the unweighted (also called macro) average sensitivity, specificity, and balanced accuracy as well as overall accuracy ([true positives +true negatives]/[true positives + true negatives + false positives + false negatives]) were calculated. These metrics are commonly used in the public health literature. However, as the current study will be of interest to the machine learning community, precision (true positives/[true positives + false positives]) and F1 scores (2 × precision × sensitivity/[precision + sensitivity]) were additionally calculated, both for each posture and movement and as unweighted averaged, as these are commonly used metrics for evaluating machine learning models [35].

*True positives* were ActiMotus classifications that were correctly classified as the posture or movement of interest (according to video coding) and *false positives* were classifications that were incorrectly classified as a posture or movement. *True negatives* were the ActiMotus classifications that were correctly classified as not belonging to the posture or movement of interest, whereas *false negatives* were the ActiMotus classifications incorrectly classified as not belonging to the posture or movement of interest.

As a secondary analysis, performance of the ActiMotus software was compared between sex and age groups using separate one-way ANOVAs to test mean differences in F1 scores and balanced accuracy for each of the postures and movements. Scheffe’s test was used for post hoc pairwise comparisons of balanced accuracy between age groups. A critical alpha probability level was set at *p* < 0.05 for all analyses. All analyses were conducted in R [36], using the “caret” [37], “car” [38] and “DescTools” [39] packages.

## 3. Results

### 3.1. Participants and Video Coding Descriptives

A total of 48 children were recruited (21 boys and 27 girls), with a mean age of 8.5 years (SD = 3.3; boys 8.7 (3.5) and girls 8.2 (3.2) years) (Table 2). A total of 5,059,345 video frames of 168,929 s duration provided 7799 instances of coded postures and movements for this analysis. Table 3 shows the video coding distribution of total seconds, total video frames, total number of instances (i.e., individual contiguous periods of a given posture or movement), mean video, and mean seconds per instance. *Sitting* and *standing* were the most frequently coded postures.

### 3.2. Agreement Between ActiMotus and Video Coding Classification

#### 3.2.1. Whole Sample Classification

Figure 1 shows the confusion matrix for the whole sample (n = 48), with the bottom left to top right representing where ActiMotus correctly classified the video coded postures and movements (i.e., true positives, marked with borders). *Lying* was correctly classified 81.0% of the time, with 18.0% being misclassified as *sitting*. Similarly, *sitting* was correctly classified 71.0% of the time, with 27.0% being misclassified as *lying*. *Standing* and *walking* were often classified as *moving* (22.0% and 23.0%, respectively). ActiMotus correctly classified *running* 64.0% of the time, with *walking* and *moving* being the most common incorrect classifications. *Stair climbing* was only correctly classified 25.0% of the time, with the most common incorrect classifications being *running* and *moving*.

Table 4 shows the comparison between ActiMotus and video coding classifications by summarising the sensitivity (called recall in the ML community), specificity, precision, balanced accuracy and F1 score for each posture and movement as well as the overall unweighted average. The overall accuracy was 74.4%. The balanced accuracy was the highest for *lying*, *sitting*, *standing*, *walking* and *running*, which ranged between 81.9 and 88.2% and lowest for *stair climbing*, which was 65.7%. The commonly used evaluation metrics when using machine learning models, F1 score, was generally lower than the balanced accuracy and the lowest for stair climbing (37.2). The overall accuracy was 74.4%.

For different postures and movements, as shown in the confusion matrix, the sensitivity was lower for *sitting* compared to *lying*. However, the opposite was found for specificity and precision, which were lower for *lying* compared to *sitting.* As a result, while balanced accuracy was similar for both *lying* and *sitting* (82.7% and 82.2%, respectively), the F1 score was lower for *lying* compared to *sitting* (63.2% and 79.9%, respectively). *Walking* had poorer sensitivity and precision, being often misclassified as *moving*, as seen in the confusion matrix. *Running* had higher balanced accuracy but a lower F1 score than *walking*. Of all the postures and movements, *stair climbing* had the lowest balanced accuracy and F1 score of 65.7% and 37.2%, respectively, due to low sensitivity (31.9%) and moderate precision (45.5%) from misclassifications to *running* and *moving* in particular.

#### 3.2.2. Sex-Stratified Classification

Figure 2 shows the confusion matrices stratified by sex. As with the total sample, for boys and girls, *lying* and *sitting* were correctly classified most of the time (>70%) and *stair climbing* was only correctly classified 24–25% of the time.

For overall postures and movements, there were no statistically significant differences between boys and girls in balanced accuracy or F1 scores (Table 5). For each posture and movement, both boys and girls had the highest F1 scores for *lying*, *sitting*, and *standing* (62.1−85.1%), while the lowest F1 scores were found for *running* among girls (45.2%) and *stair climbing* among boys (41.6%). The highest balanced accuracy was found for *lying*, *sitting*, *standing*, and *running* (between 79.1 and 84.2%). Girls had a significantly higher balanced accuracy (71.6%) compared to boys (64.8%) for *stair climbing*.

#### 3.2.3. Age-Stratified Classification

When assessing the confusion matrices for the different age groups (Figure 3), there was a similar pattern of misclassification across age groups; however, agreement between the ActiMotus and video-classified posture and movements was poorest among children aged 3–5 years old compared to the older children. For example, *running* was only correctly classified 61.0% of the time in this group of younger children compared to 64.9% and 74.0% of the time for the children aged 6–10 and 11–14 years, respectively.

When considering all postures and movements, the older age group had a higher accuracy as measured with F1 scores compared to the youngest age group (Table 5). When considering specific postures and movements, a significantly higher accuracy both in terms of F1 scores and balanced accuracy for *walking*, *running*, and *stair climbing* among the older age group was observed.

## 4. Discussion

This study evaluated the criterion validity of the ActiMotus software to classify postures and movements among children between 3 and 14 years old in a laboratory setting. Overall, we found that ActiMotus classified children’s posture and movements with a level of accuracy (with an F1 score of 61.9% and a balanced accuracy of 81.1%) comparable to the levels reported among older children [24]. The level of accuracy differed between the postures and movements assessed. Based on balanced accuracy, ActiMotus classified lying, sitting, standing, walking, and running with a high level of accuracy (≥82%), while stair climbing showed a lower level of accuracy of 65.7%. Based on F1 scores, the highest levels of accuracy were found for lying, sitting, and standing (63.2–85.3%), while walking, running, and stair climbing had the lowest levels of accuracy (37.2–57.7%). We found higher balanced accuracy for stair climbing among girls compared to boys, and a higher accuracy (both in terms of F1 scores and balanced accuracy) for the older age group (11–14 years) for the more dynamic movements, i.e., walking, running, and stair climbing.

ActiMotus could classify lying and sitting with a level of balanced accuracy similar to what has been reported in studies based on hip-worn accelerometry using different classification software [40,41]. To our knowledge, only two studies have evaluated the accuracy of thigh-worn accelerometers to capture lying and sitting separately among children [21,24]. The laboratory-based study by Stewart et al. used a thigh-worn Axivity AX3 accelerometer and custom-made machine learning software among children aged 7–15 years [21]. The authors found slightly lower balanced accuracy for lying (77%) but a higher level of balanced accuracy for sitting (87.8%) than what we observed in the current study (i.e., lying 82.7% and sitting 82.2%). The study by Lendt et al. was based on the same laboratory-based dataset but supplemented this with thigh-worn free-living accelerometer data from 15 children, aged 8–12 years and processed the data using a similar algorithm as the one in the current study as well as the modified version published by Brønd et al. [24]. The authors reported a lower balanced accuracy for lying across conditions and algorithms (61–62%) than what we observed in the current study but similar levels of accuracy as we observed were found for standing, walking and running. The differences found for lying and sitting between our study and that of Stewert et al. might be because the ActiMotus algorithm only considers ≤10 accelerometer signal features and is therefore ”simpler” than the complex machine learning classification algorithm, which considers more than 100 features [21]. The difference in accuracy could also be a result of differences in the ages of the children included, as both Stewart et al. and Lendt et al. did not consider children under the age of 7 years [21,24]. It should also be highlighted that to improve ecological validity, the children in the current study performed different lying positions and were free to choose their preferred sitting position, thus varying the position of the thigh (e.g., prone, side and supine lying, cross-legged, or long sitting). Thus, our finding suggests that ActiMotus can accurately measure lying and sitting with varied thigh positions in children.

Although ActiMotus classified standing, walking, and running with a high level of balanced accuracy (82–88%), we did observe that the software sometimes classified these behaviours as “moving”. The “moving” ActiMotus classification was created to capture times where adults were neither standing still nor walking but had slight movement of the thigh while being in an upright position. This category was not coded in the current study as the human coding of this classification was considered not sufficiently reliable during pilot coding. This was contributed to by the camera being too far away to accurately detect that movement was occurring and that the coding of the video recording was conducted frame by frame, which hindered coding of short bursts of fidgeting movements. Thus, some of the data coded as standing or walking by the humans could have been correctly classified as “moving” by ActiMotus. If so, this would suggest that the detection of small movements may be more feasible using accelerometers compared to direct observations among children. The classification of running as moving or walking may be a result of the sampling frequency of 12.5 Hz used for this study. Compared to higher sampling frequencies, the signal of the raw accelerometer data sampled at 12.5 Hz is smoother, resulting in fewer and smaller peaks for faster movements. This influences the standard deviation of the sliding windows, which is used in the ActiMotus algorithm, potentially resulting in decreased ability to accurately classify running. Further work is currently underway to refine the algorithm in a number of ways, including examining the impact of using data sampled at higher frequencies.

We found comparable levels of balanced accuracy with what has previously been reported by validation studies using thigh-worn accelerometry and custom-made software to measure standing, walking, and running among children [20,21,22,23]. The studies by Stewart et al. and Lendt et al. found slightly higher levels of balanced accuracy for standing, walking, and running (>86%) than the current study [21,24]. Likewise, the study by Brønd et al. found higher levels of balanced accuracy for standing, walking, and running (>90%) among children aged 3–16 years old [20] than what we observed. In contrast, two studies using a thigh-worn activPAL accelerometer and the activPAL software to measure standing and walking among children aged 4–6 years found lower or comparable levels of balanced accuracy for classifying standing (76–89%) and walking (74–88%) [22,23]. Although differences in hardware, software, and study design hinder direct comparison, the comparability in balanced accuracy between ActiMotus and previous work suggests that ActiMotus performs similarly to what has previously been reported in classifying standing, walking, and running among children.

Stair climbing was often misclassified as walking and running (10% and 41% of the time, respectively), which can be expected as the children both walked and ran up and down the stairs. Furthermore, the classification of stair climbing could have been impacted by the height of the children relative to the height of the stairs, whether the child was holding the handrail, and the movement pattern (e.g., whether the child was stepping reciprocally or not). To the best of our knowledge, this is the first study to evaluate the accuracy of using accelerometry for classifying when children are stair climbing. However, the ActiMotus software has been shown to measure stair climbing with high levels of balanced accuracy among adults (98%) [25]. The discrepancies in balanced accuracy might be attributable to differences in how the stair climbing was performed. For example, some children choose to jump up the stairs, thereby skipping several steps. Thus, our results suggest that stair climbing may be harder to classify in children and highlight the challenges in capturing the highly variable movement patterns among children, which should be considered when developing sensor-based systems for classifying postures and movements among children.

Overall, ActiMotus showed comparable levels of accuracy in classifying postures and movements between boys and girls, suggesting that ActiMotus can classify postures and movements similarly between sexes; however, we found a slightly higher balanced accuracy among girls compared to boys for stair climbing. Although this result might be a consequence of multiple testing and/or a small sample size, the difference may suggest that girls and boys move differently when climbing stairs. To the best of our knowledge, no prior validation study has evaluated potential sex differences in accelerometer measurements of posture and movements, which should be investigated further. 

When investigating potential age differences, we observed lower levels of accuracy among young compared to older age groups for dynamic movements such as running and walking. Similarly, Brønd et al. found lower balanced accuracy for walking and running among pre-school children (3–6 years old) compared to older children (9–12 years old) and adolescents (13–16 years old) [20]. It is well-acknowledged that children tend to move in ways that are more varied, sporadic, and disorganised than adults [27,42]. As the ActiMotus software was developed based on data from an adult sample [25], our findings further support this and could indicate that particularly younger children move more differently than adults compared to older children. We encourage future work to examine the kinematic aspects of movements that vary between younger and older children and evaluate how decision tree algorithms could integrate such information to enhance accuracy across a wide range of children’s ages.

### 4.1. Practical Implications

Valid measurements of children’s daily postures and movements are essential for understanding how and why children move as well as the implications for health, wellbeing, and development [5]; however, obtaining such measurements in children given their complex and intermittent movement pattern has been challenging [27]. We found that ActiMotus showed high level of accuracy for lying, sitting, and standing, while the accuracy for walking and running was lower. Thus, the ActiMotus could be a suitable software for measuring some postures and movements among children. The Motus system is therefore a potentially viable method for more accurate measurement of postures and movements in small intervention studies but could also be used in large surveillance studies as recently recommended [43]. Nevertheless, the software can be improved for measuring more dynamic movement, i.e., walking, running, and stair climbing, and among younger age groups. One way this could be achieved is by refining the ActiMotus classification algorithm. Brønd et al. refined the ActiMotus classification algorithm by changing thresholds in the decision tree [20], which was further evaluated in the study by Lendt et al. [24]. Both studies reported high levels of balanced accuracy and Lendt et al. did not find any significant differences between the modified and the original ActiMotus classification algorithm. Nevertheless, the algorithm has not been refined for detecting stair climbing or more child-specific movements, e.g., skipping and crawling. It is also worth noting that the ActiMotus is based on data from a single thigh-worn accelerometer and the algorithm may be improved by adding data from a second accelerometer. Thus, we encourage more work to refine the ActiMotus algorithm for the paediatric population. Such future work could examine if a dual accelerometer setup improves the accuracy of the algorithm. Another area of future work could be to investigate whether providing data on age, sex, and anthropometric measures of the participants could enable automatic selection of an algorithm that changes parameters and thresholds used for classification of postures and movements. 

Alternatively, machine learning models as used by Stewart et al. [21] can predict postures and movements by learning complex patterns in raw accelerometer data [44], which may enable greater accuracy for measuring postures and movements among children. Work is currently underway to investigate this [45]. Future studies comparing the performance of simple algorithms to machine learning models for measuring postures and movements among children are warranted and all validation studies should report the range of accuracy metrics used to facilitate comparisons. 

We evaluated accuracy with the metrics commonly used in both public health and machine learning fields. The different measures provide important insights into the performance of classification software regarding the postures and movements that children are (by assessing sensitivity, specificity, and balanced accuracy) and are not (by precision and F1 scores) doing. Thus, we encourage future work to present and evaluate classification software using the same evaluation metrics. 

### 4.2. Strengths and Limitations

A strength of the current study was the inclusion of children across multiple age groups (between 3 and 14 years old). Another strength was the use of the ActiMotus software, which is based on a decision tree using simple biomechanical features and is publicly available, allowing researchers to use, replicate, and potentially refine the algorithm. Although the children followed a structured activity protocol, several adjustments to the protocol were made to ensure that the children moved as they would outside a laboratory. Specifically, all activities were age-appropriate and children performed each activity in varied positions. Nevertheless, the data were collected under semi-controlled settings in a laboratory and thus the observed accuracy of ActiMotus could reduce when used under free-living conditions such as running on varied surfaces and sitting in a vibrating car. We encourage future work to consider free-living validation studies of ActiMotus in a range of age groups. Another limitation of our study design was the video recording in the laboratory. The camera was set up 4–6 m away from the centre of the laboratory to ensure that the whole setting was captured and limit reactivity among the children. However, this may have resulted in less accurate human coding as it was sometimes challenging to see which posture or movement the children were performing if they were far away from the camera. In line with this limitation, we identified some issues with the human coding of some postures and movements, for example separating standing from moving. Furthermore, synchronisation between the video and accelerometer time stamps was challenging due to clock differences. These two issues may have resulted in some of the identified misclassifications of the ActiMotus software. Finally, we used a sampling frequency of 12.5 Hz, which may have been too low to optimally classify higher-intensity activities, such as running.

## 5. Conclusions

ActiMotus, a classification software based on thigh-worn accelerometry, showed high levels of accuracy for detecting lying, sitting, and standing among children, while lower levels of accuracy were found for classifying walking, running, and stair climbing. Further, we observed higher balanced accuracy among girls compared to boys for stair climbing and that the software performed better for older compared to younger children. Thus, the software could be improved for classifying the more dynamic postures and movements, particularly among younger children. We encourage future studies to assess if such improvements could be achieved by refining the ActiMotus algorithm or if more advanced methods, such as machine learning models, are required. Further, more work is needed to develop a software classification feasible for measuring other postures and movements that children may commonly perform.

## Figures and Tables

**Figure 1 sensors-24-06705-f001:**
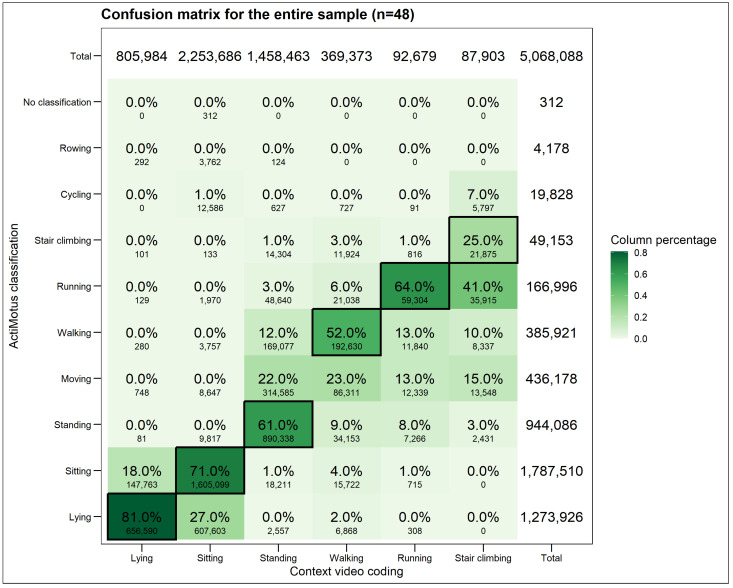
Confusion matrix for ActiMotus and video coded postures and movements based on the whole sample (N = 48).

**Figure 2 sensors-24-06705-f002:**
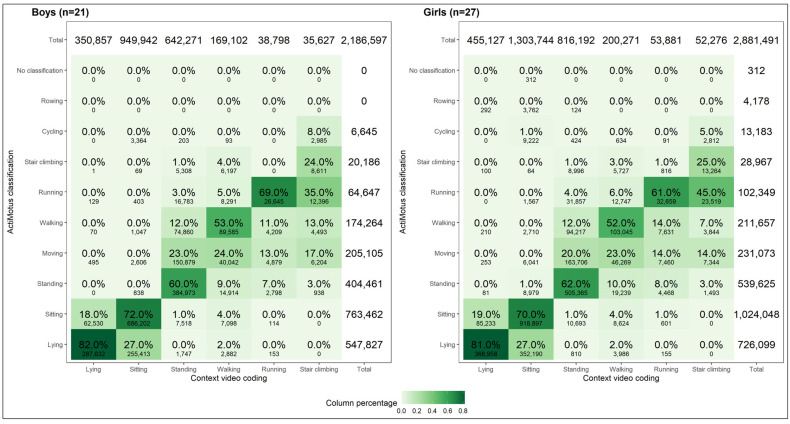
Confusion matrix for ActiMotus and video coded postures and movements stratified by sex.

**Figure 3 sensors-24-06705-f003:**
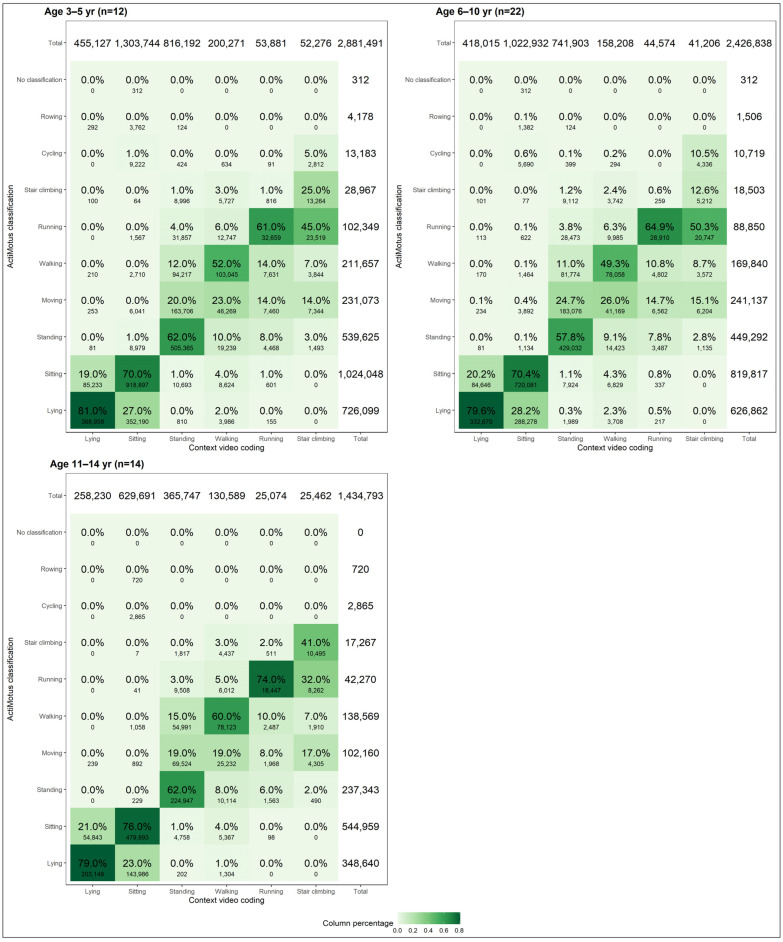
Confusion matrices for ActiMotus and video coded postures and movements stratified by age group.

**Table 1 sensors-24-06705-t001:** Definition of each posture and movement used for coding of the videos.

Posture/Movement Coding	Description
Lying prone	Child lying prone on any surface, knees can be flexed or extended.
Lying supine	Child lying supine on any surface, knees and hips and can be flexed or extended.
Lying side	Child lying on right or left side. Must truly be on side otherwise classify as prone or supine.
Sitting	Child sitting on any surface (floor, couch, beanbag, chair, stool, swivel chair) with any leg posture (including “W sitting”, cross-legged sitting, legs out straight, etc).
Low kneeling	Child sitting on knees with bottom on heels.
High kneeling	Child on knees with bottom off heels.
Half kneeling	Child on one knee with other foot on floor, bottom off heels.
Standing	Child standing, can be moving or can be stationary—note that this was selected as children frequently fidgeted or wriggled while standing so differentiating between standing with movement and standing still was not feasible.
Throwing and catching ball	Child throwing or catching a ball.
Walking	Child walking at any speed, to be classified as walking there needed to be a full gait cycle present (i.e., 2 steps).
Running	Child running at any speed.
Skipping	Child skipping or variations in skipping due to differences in motor development (e.g., some younger children when instructed to skip used a galloping action).
Jumping	Child jumps continuously either on the spot or travelling forward or jumps off an object (e.g., jumps off stairs).
Walking upstairs	Child walks upstairs.
Walking downstairs	Child walks downstairs.
Running upstairs	Child runs upstairs.
Running downstairs	Child runs downstairs.
Handstand	Child attempts a handstand. Dependent upon proficiency and motor development this may be placing their hands on the ground and kicking feet off ground or may be a sustained handstand.
Cartwheel	Child attempts a cartwheel. As for handstand this may be placing hands on ground and kicking feet off one at a time.
All fours	Includes crawling, bear walking.
Transition	Includes all transitional movements including sit to stand, stand to sit, lie to stand, stand to lie, rolling over.
Other	Any movement that is not captured above.
Uncodable	The child is outside of the view of the ground truth video recording or could not be confidently annotated due to image quality.
Unsure	The coder is uncertain of what to code.

**Table 2 sensors-24-06705-t002:** Age distribution of study participants by sex.

	Girls	Boys
Age Categories	(n; Mean (SD))	(n; Mean (SD))
3–6 years	7; 4.1 (0.9)	5; 3.8 (0.8)
7–10 years	11; 8.4 (1.4)	11; 8.4 (1.3)
11–14 years	9; 12.6 (1.1)	5; 12.4 (1.1)
Total	27	21

**Table 3 sensors-24-06705-t003:** Sum of video coded data across all participants and standard deviation of individual participant data.

	Lying	Sitting	Standing	Walking	Running	Stair Climbing	Overall
Total seconds (SD)	26,863 (219)	75,109 (262)	48,649 (372)	12,285 (104)	3096 (26)	2927 (30)	168,929 (588)
Total video frames (SD)	805,563 (6559)	2,252,724 (7940)	1,454,737 (11,154)	366,862 (3107)	92,175 (777)	87,284 (884)	5,059,345 (17,620)
Total instances (SD)	410 (4)	855 (11)	3145 (31)	2293 (25)	480 (6)	616 (7)	7799 (28)
Mean video frames per instance (SD)	2157 (917)	3144 (1074)	520 (221)	173 (55)	236 (126)	148 (49)	1075 (1309)
Mean seconds per instance (SD)	72 (31)	105 (36)	17 (7)	6 (4)	8 (4)	5 (2)	35 (44)

Standard deviation (SD) is calculated based on individual participant data, which are summed to provide total seconds, video frames, and instances.

**Table 4 sensors-24-06705-t004:** Comparison between ActiMotus and video coding classification for the whole sample (N = 48).

	Lying	Sitting	Standing	Walking	Running	Stair Climbing	Overall
Sensitivity (%)	81.6	72.0	77.9	68.2	73.9	31.9	67.6
Specificity (%)	83.8	92.3	98.4	95.5	97.6	99.4	94.5
Precision (%)	51.5	98.8	94.3	49.9	35.5	45.5	62.6
F1 score (%)	63.2	79.9	85.3	57.7	48.0	37.2	61.9
Balanced accuracy (%)	82.7	82.2	88.2	81.9	85.8	65.7	81.1

**Table 5 sensors-24-06705-t005:** Comparison between ActiMotus and video coding classification stratified by sex and age groups.

	Sex				Age Group (Years)				
	Boys	Girls	F (df1, df2)	*p*	3–5	6–10	11–14	F (df1, df2)	*p*
F1 scores (%)									
Lying	66.9	62.1	0.7 (1, 44)	0.419	58.7 ^a^	65.4 ^a^	67.4 ^a^	0.9 (2, 43)	0.412
Sitting	77.1	77.0	>0.1 (1, 46)	0.988	77.3 ^a^	74.4 ^a^	80.9 ^a^	0.5 (2, 45)	0.610
Standing	85.1	82.9	0.1 (1, 46)	0.343	83.4 ^a^	85.0 ^a^	82.4 ^a^	0.5 (2, 45)	0.636
Walking	58.8	54.7	1.5 (1, 46)	0.228	51.9 ^a^	54.9 ^a,b^	63.0 ^b^	3.8 (2, 45)	0.029
Running	52.4	45.2	2.2 (1, 45)	0.147	41.1 ^a^	46.8 ^a,b^	57.1 ^b^	3.4 (2, 44)	0.041
Stair climbing	41.6	54.7	3.2 (1, 28)	0.085	46.0 ^a,b^	32.4 ^a^	64.3 ^a,b^	12.4 (2, 27)	>0.001
Overall	64.4	63.7	>0.1 (1, 46)	0.841	60.7 ^a,b^	62.2 ^a^	69.4 ^b^	3.9 (2, 45)	0.028
Balanced accuracy (%)									
Lying	84.2	80.5	1.5 (1, 44)	0.270	83.3 ^a^	81.1 ^a^	82.9 ^a^	0.2 (2, 43)	0.852
Sitting	82.7	82.7	>0.1 (1, 46)	0.847	81.5 ^a^	81.4 ^a^	84.5 ^a^	0.3 (2, 45)	0.728
Standing	79.1	79.6	0.1 (1, 46)	0.786	80.9 ^a^	78.6 ^a^	79.3 ^a^	0.6 (2, 45)	0.534
Walking	75.3	72.6	2.0 (1, 46)	0.159	71.2 ^a^	72.6 ^a,b^	77.7 ^b^	4.3 (2, 45)	0.020
Running	81.6	79.7	0.5 (1, 45)	0.493	72.6 ^a^	80.9 ^a,b^	86.9 ^b^	9.8 (2, 44)	>0.001
Stair climbing	64.8	71.6	4.2 (1, 28)	0.049	67.0 ^a,b^	61.5 ^a^	75.1 ^b^	9.4 (2, 27)	0.001
Overall	81.9	80.8	0.3 (1, 46)	0.606	81.1	80.1	83.1	4.3 (2, 45)	0.470

F(df1, df2) = F statistic on degrees of freedom (numerator) and degrees of freedom (denominator), respectively. *p* = value based on one-way ANOVA results. ^a,b^ Scheffe’s pairwise comparison of F1 score and balanced accuracy, respectively. If two groups share a letter, they are not significantly different.

## Data Availability

The datasets collected and analysed during the current study are not publicly available due to ethical constraints.

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
