# Peer review of "Evaluation of the ActiMotus Software to Accurately Classify Postures and Movements in Children Aged 3–14"

_sensors, 2024, doi:10.3390/s24206705_

Round 1

Reviewer 1 Report

Comments and Suggestions for Authors

The authors describe and discuss the results of a work on the determination of postures and movements of 3-14 aged children using the Motus wearable  ensor, that has proved high accuracy in that task among adults. This study also compares results in three different ranges of age and between sexes. The sensor used was an accelerometer attached to the children thigh. Posture classification obtained from the accelerometer was checked against  videorecordings using two smartphones placed on either side of the laboratory for a better determination of the children movements, while being unobtrusive for the participants.

The experiment is well described, conditions of measurements, selection of participants, structure of data collection sessions, and data recording, both with the sensors and in video. The analysis, synchronization and comparison of both types of data, accelerometer and video, is also well described, as well as the validity of the human codification of movements in the videorecordings. Perhaps the frequency of the accelerometer, 12.5 Hz, wasn't high enough and this is a possible reason of some errors in the determination of movements, as the authors discuss later in the paper.

Confusion matrices with the misclassification by the sensor Motus of each movement are discussed by comparison with the human classification of the videorecordings. Different parameters, sensitivity, specificity, accuracy, ... are used to quantify the accuracy of Motus classification. As a second result, data from Motus are also used to statistically analyse differences in Motus accuracy depending on the age and sex of the participants. All the results are discussed by the authors, and possible reasons of the disagreements, between Motus and video, and differences, due to age or sex, are also analyzed.

Perhaps, one of the points of improvement is the use of a low frequency for the accelerometer measurements. If this work is planned as a possible first step towards an automated or AI detection system, the increase in that frequency to a higher, though reasonable frequency, shouldn't be an issue. 

From my point of view, some of the results in the confusion matrices are reasonable, even too good, considering that only one sensor (accelerometer) was used. For example, how can Motus, from only one sensor attached to the thigh classify a posture as lying or sitting correctly? I can guess similar  difficulties in other classifications shown in the confusion matrices. Then, apart from the discussion on the accuracy of the results that is done in the paper, mainly from the experimental sources of error, both the sensor characteristics and the different behavior/attributes of the subjects, I think it could be useful if the authors could add a brief discussion on how these results show weak points of the classification algorithm, and/or if this work could also be useful for its improvement, beyond the measurement, determination and description of its good or not so good results. Could it be possible to have a brief discussion on this? Perhaps some of this is mentioned in page 12 , when Motus' algorithm results are compared with Stewert et al. results, or in page 14, but more (I think) as a possible source of confusion than as a hint on improvement lines. Of course, this is not the aim of the experiment and paper, but it could also be useful if the authors, as users that try to evaluate Motus classification, could briefly discuss also in that direction.

Author Response

Comment #1: The authors describe and discuss the results of a work on the determination of postures and movements of 3-14 aged children using the Motus wearable sensor, that has proved high accuracy in that task among adults. This study also compares results in three different ranges of age and between sexes. The sensor used was an accelerometer attached to the children’s thigh. Posture classification obtained from the accelerometer was checked against videorecordings using two smartphones placed on either side of the laboratory for a better determination of the children movements, while being unobtrusive for the participants.

The experiment is well described, conditions of measurements, selection of participants, structure of data collection sessions, and data recording, both with the sensors and in video. The analysis, synchronization and comparison of both types of data, accelerometer and video, is also well described, as well as the validity of the human codification of movements in the videorecordings. Perhaps the frequency of the accelerometer, 12.5 Hz, wasn't high enough and this is a possible reason of some errors in the determination of movements, as the authors discuss later in the paper.

Confusion matrices with the misclassification by the sensor Motus of each movement are discussed by comparison with the human classification of the videorecordings. Different parameters, sensitivity, specificity, accuracy, ... are used to quantify the accuracy of Motus classification. As a second result, data from Motus are also used to statistically analyse differences in Motus accuracy depending on the age and sex of the participants. All the results are discussed by the authors, and possible reasons of the disagreements, between Motus and video, and differences, due to age or sex, are also analyzed.

Author reply:
We thank the reviewer for the positive review. Please find our point-by-point replies to the issues raised below.

We would further like to inform you the research team behind the Motus system has finalised their branding since submitting this manuscript. Specifically, the Motus software is now called ActiMotus.  Thus, we have replaced the name “Motus” with “ActiMotus” throughout the manuscript and changed the manuscript title accordingly.

Comment #2: Perhaps, one of the points of improvement is the use of a low frequency for the accelerometer measurements. If this work is planned as a possible first step towards an automated or AI detection system, the increase in that frequency to a higher, though reasonable frequency, shouldn't be an issue. 

Author reply:
We agree with the review that the ‘low’ frequency of the accelerometer data may account for some of the poorer classification accuracy. There is indeed current work underway to examine this. In the previous version of the manuscript, we wrote:

“The classification of running as moving or walking may be a result of the sampling frequency of 12.5Hz used for this study. Compared to higher sampling frequencies, the signal of the raw accelerometer data sampled at 12.5Hz is smoother, resulting in fewer and smaller peaks for faster movements. This influences the standard deviation of the sliding windows, which is used in the ActiMotus algorithm, potentially resulting in decreased ability to accurately classify running. Further work is currently underway to refine the algorithm to take this into account.”

We have edited the sentence about further work to make it more explicit. Thus, the last sentence in this paragraph now reads:

“Further work is currently underway to refine the algorithm in a number of ways, including examining the impact of using data sampled at higher frequencies.”

Comment#3: From my point of view, some of the results in the confusion matrices are reasonable, even too good, considering that only one sensor (accelerometer) was used. For example, how can Motus, from only one sensor attached to the thigh classify a posture as lying or sitting correctly? I can guess similar difficulties in other classifications shown in the confusion matrices.

Author reply:

We agree that using a single accelerometer placed on the thigh will have limitations to capture a range of postures and particularly those where the positions of the thigh are very similar such as lying and sitting. However, the Motus algorithm has been shown to classify a range of postures and movements with an even higher level of accuracy among adults than those reported in our study on children. This is achieved as the algorithm considers three main features – the inclination of the thigh, the angle of the thigh, and the standard deviation of three accelerometer axis. Furthermore, the algorithm has been developed to distinguish between sitting and lying based on a feature in the algorithm where the rotation of the thigh during a sedentary period is considered, as lateral rotation of the thigh is rare in sitting but common when lying.

That said, it may be that the algorithm’s classification accuracy would be improved if using data from a dual accelerometer setup (e.g. thigh combined with wrist or trunk). To our knowledge, only the referred study by Stewart et al. has investigated this among children and thus, we encourage further work to investigate this.

We have added details about how the algorithm distinguishes between sitting and lying in the method section, which now reads:

“A custom developed software, ActiMotus, was used to classify postures and movements based on 12.5Hz data from the accelerometers. The ActiMotus software is based on the previously developed Acti4 [25] and ActiPASS software [32] and is publicly available at https://github.com/motus-nfa/ActiMotus [33]. ActiMotus uses accelerometer data resampled to 30Hz in 2 second windows with 50% overlap to classify different postures and movements based on a rule-based algorithm, which considers the inclination of the thigh, the angle of the thigh, the standard deviation of thigh accelerations and the rotation of the thigh [25] [32].”

Furthermore, we have the following sentence in the Discussion suggesting further work using data from dual accelerometer setup:

Nevertheless, the algorithm has not been refined for detecting stair climbing or more child-specific movements, e.g. skipping and crawling. It is also worth noting that the Ac-tiMotus is based on data from a single thigh-worn accelerometer and the algorithm may be improved by added data from second accelerometer. Thus, we encourage more work to refine the ActiMotus algorithm for the paediatric population. Such future work could ex-amine if a dual accelerometer setup improves the accuracy of the algorithm.”

Comment#4: Then, apart from the discussion on the accuracy of the results that is done in the paper, mainly from the experimental sources of error, both the sensor characteristics and the different behavior/attributes of the subjects, I think it could be useful if the authors could add a brief discussion on how these results show weak points of the classification algorithm, and/or if this work could also be useful for its improvement, beyond the measurement, determination and description of its good or not so good results. Could it be possible to have a brief discussion on this? Perhaps some of this is mentioned in page 12 , when Motus' algorithm results are compared with Stewert et al. results, or in page 14, but more (I think) as a possible source of confusion than as a hint on improvement lines. Of course, this is not the aim of the experiment and paper, but it could also be useful if the authors, as users that try to evaluate Motus classification, could briefly discuss also in that direction.

Author reply:

We agree with the comment and have thus added a more detailed discussion about how the algorithm could be improved on page 14. The section now reads:

“Thus, we encourage more work to refine the ActiMotus algorithm for the paediatric population. Such future work could examine if a dual accelerometer setup improves the accuracy of the algorithm. Another area of future work could be to investigate whether providing data on age, sex and anthropometric measures of the participants could enable automatic selection of an algorithm that changes parameters and thresholds used for classification of postures and movements."

Reviewer 2 Report

Comments and Suggestions for Authors

The manuscript reports on the monitoring of postures of young children with commercially available accelerometer Motus, and classification algorithm to identify varying postures. The experimental design and results reveal good level of accuracy and assessment for different postures. Besides, this paper is well-organized and written. Based on the above consideration, I think this manuscript can be accepted upon minor revision.

1.     In the introduction section, the background and significance of the study could be more explicitly elaborated, along with the reasons for choosing the Motus system for the research.

2.     How can the sensor be used to provide information on specific postures?    

3.     What are the effects of varying postures on the accuracy of wearing device?

4.     Please explain why there are differences of accuracy among various ages of children.

5.     The sampling frequency of 12.5Hz is too low to obtain optimum results, then is there any measures taken to improve this point?

Author Response

Comment #1: The manuscript reports on the monitoring of postures of young children with commercially available accelerometer Motus, and classification algorithm to identify varying postures. The experimental design and results reveal good level of accuracy and assessment for different postures. Besides, this paper is well-organized and written. Based on the above consideration, I think this manuscript can be accepted upon minor revision.

Author reply:
We thank the reviewer for the positive review. Please find our point-by-point replies to the issues raised below. We would further like to inform you the research team behind the Motus system has finalised their branding since submitting this manuscript. Specifically, the Motus software is now called ActiMotus.  Thus, we have replaced the name “Motus” with “ActiMotus” throughout the manuscript and changed the manuscript title accordingly.

Comment#2: In the introduction section, the background and significance of the study could be more explicitly elaborated, along with the reasons for choosing the Motus system for the research.

Author reply:
We agree with the comment and have revised the introduction to include a section that describes the need for valid measurements of postures and movement among children and how the Motus system has the potential to be a suitable measurement system for obtaining robust measures of postures/movements among children.

The section reads:

”Thus, there is further need to develop and validate wearable sensor based systems enabling measurements of a range of postures and movements among children, including young children. Obtaining these robust measurements are essential to enable evidence-based guidance to encourage appropriate physical activity in children to support their health and development. The Motus system was specifically designed to ensure ease of use and low burden for both researchers and participants and thus, has the potential to be widely applicable to small and large scale intervention and observations studies. While evidence shows this system does provide a robust and highly useful system among adults, it’s capabilities for use among children is not yet clear. Accordingly, this study aimed to evaluate the criterion validity of ActiMotus, the software used in Motus, to classify postures and movements among children between 3-14 years in a laboratory setting by comparing to human-coded video. We further assessed if the sex or age of children impacted the accuracy.

Comment#3: How can the sensor be used to provide information on specific postures?

Author reply:

The thigh-worn sensor provides acceleration data from which a number of parameters are calculated and used by the classification software to estimate likely posture and movement. We have now added details in the Methods section 2.5 to make explicit the parameters used. The section now reads:

“A custom developed software, ActiMotus, was used to classify postures and movements based on 12.5Hz data from the accelerometers. The ActiMotus software is based on the previously developed Acti4 [25] and ActiPASS software [32] and is publicly available at https://github.com/motus-nfa/ActiMotus [33]. ActiMotus uses accelerometer data resampled to 30Hz in 2 second windows with 50% overlap to classify different postures and movements based on a rule-based algorithm, which considers the inclination of the thigh, the angle of the thigh, the standard deviation of thigh accelerations and the rotation of the thigh [25] [32].”

Comment#4: What are the effects of varying postures on the accuracy of wearing device?

Author reply:

We believe the reviewer is asking if it is possible that the sensor can be knocked or moved by various positions/movements and how this might impact the accuracy of the algorithm. Given that the sensor is applied to the thigh with medical grade tape it did not move relative to the thigh during our data collection, in any of the positions or movements performed.

Comment#5: Please explain why there are differences of accuracy among various ages of children.

Author reply:

We found that the algorithm had lower levels of accuracy among younger compared to older age-groups, particularly for the dynamic postures and movements, i.e. walking, running and stair climbing. As children age, significant changes of the skeleton and muscle mass occurs, which influences the biomechanics of how children move. For example, it can be expected that younger children have shorter stride lengths when walking and running compared to older children. Given that the algorithm was developed based on accelerometer data from adults, we can therefore expect a greater classification accuracy of the algorithm among older children compared to younger children, which our results showed.

We have revised the paragraph in the discussion section to include a statement which encourages further research to examine this topic, which now reads:

It is well-acknowledged that children tend to move in ways that are more varied, sporadic, and disorganised than adults [27,42]. As the ActiMotus software was developed based on data from an adult sample [25], our findings further support this and could indicate that particularly younger children move more differently than adults compared to older children. We encourage future work to examine the kinematic aspects of movements that vary between younger and older children and evaluate how decision tree algorithms could integrate such information to enhance accuracy across a wide range of children’s ages.”

Comment#6: The sampling frequency of 12.5Hz is too low to obtain optimum results, then is there any measures taken to improve this point?

Author reply:

We agree with the reviewer that the 12.5hz frequency may be limitation and there is current work underway to investigate the potential impact of difference data sampling frequencies on the accuracy of the algorithm.

In the previous version of the manuscript, we wrote:

“The classification of running as moving or walking may be a result of the sampling frequency of 12.5Hz used for this study. Compared to higher sampling frequencies, the signal of the raw accelerometer data sampled at 12.5Hz is smoother, resulting in fewer and smaller peaks for faster movements. This influences the standard deviation of the sliding windows, which is used in the ActiMotus algorithm, potentially resulting in decreased ability to accurately classify running. Further work is currently underway to refine the algorithm to take this into account.”

We have edited the sentence about further work to make it more explicit. Thus, the last sentence in this paragraph now reads:

“Further work is currently underway to refine the algorithm in a number of ways, including examining the impact of using data sampled at higher frequencies.”